# Application of Thermochemical Method to Determine the Crystallinity Degree of Cellulose Materials

## Michael Ioelovich

Designer Energy, Rehovot 7670504, Israel; ioelovichm@gmail.com; Tel.: +972-89366612

**Abstract:** Currently, to characterize the crystallinity of cellulose, such an estimated parameter as the crystallinity index is used, measured by various methods and techniques. The main purpose of this article was to develop a thermochemical method for determining the real degree of crystallinity (X) of cellulose based on the measurement of the enthalpy of wetting. Various cellulose samples, such as MCC, pure cotton cellulose, bleached wood pulps, mercerized celluloses, and viscose rayon fibers, were used. For these samples, the exothermic wetting enthalpy ($\Delta H_w$), the maximum amount of sorbed moisture ($A_o$), as well as the X-ray index of crystallinity (CrI) were studied. The dependence of $\Delta H_w$ on $A_o$ was linear and can be expressed by the equation: $\Delta H_w = k\,A_o$, where the coefficient $k = -336$ (J/g). After substituting the theoretical value $A_{o,a} = 0.5$ (g/g) into this equation, the numerical value of maximum wetting enthalpy $\Delta H_{w,a} = -168$ (J/g) for completely amorphous cellulose was obtained. As a result, the equation for calculating the real crystallinity degree (X) expressed in mass fractions was derived: $X = 1 - (\Delta H_w / \Delta H_{w,a})$. Analysis of the obtained results showed that only the X parameter can characterize the real content of crystallites in cellulose samples, instead of the approximate CrI parameter.

**Keywords:** cellulose; thermochemical method; wetting enthalpy; degree of crystallinity; calculations





## 1. Introduction

Being a linear stereoregular semicrystalline polysaccharide, cellulose is the most abundant natural organic matter on Earth [1]. This biopolymer is present in all terrestrial plants, algae, and tunicates; furthermore, it is also synthesized by some microorganisms [2,3]. To isolate chemically pure cellulose from natural sources, two main industrial pulping methods are applied, such as sulfite and kraft processes accompanied by multi-stage bleaching [4,5]. Several pilot and laboratory pulping methods also exist, e.g., soda, oxidative, Organosolv, etc. [6,7]. The resulting matter isolated from natural biomasses is cellulose, a semicrystalline biopolymer having crystalline allomorph CI (α and β). The crystallinity of isolated cellulose depends on the used natural source and the isolation conditions [8]. In addition, the physicomechanical treatments (e.g., ball-grinding [9]), physicochemical modifications (e.g., treatment with liquid ammonia, primary amines, and diamines, some solvents, and concentrated solutions of alkalis [8,10–12]), and chemical modifications (e.g., hydrolysis with boiling dilute mineral acids [3,8]) of natural cellulose lead to a significant change in the crystallinity.

In the subsequent structural analysis of cellulose samples, a problem arises due to the lack of an accurate method for determining such an important characteristic as the degree of crystallinity. In fact, there are only approximate indices of cellulose crystallinity, which can be estimated by various methods, such as wide-angle X-ray scattering (WAXS), solid-state $^{13}$C NMR, FTIR, Raman spectroscopy, as well as by physicochemical (e.g., sorption iodine), chemical (e.g., hydrolysis with boiling 2.5 M HCl) methods, etc. [3,13–18]. However, different methods give different values of the crystallinity index (CrI) for the same cellulose sample; therefore, it is not clear which value of CrI should be preferred.

Even applying the same method, e.g., WAXS, also gave different CrI values for the same sample if different measurement techniques were used to separate X-ray scattering

from the crystalline and amorphous domains. So, the study of microcrystalline cellulose Avicel PH-101 showed that the measuring of peak heights gave CrI 0.80–0.93, deconvolution of the peaks gave CrI 0.55–0.61, subtraction of amorphous scattering gave CrI 0.60–0.78, Ruland's technique gave CrI 0.55–0.61, the technique of Jayme–Knolle gave CrI about 0.69, and the technique of Hermans–Weidinger gave CrI ranging from 0.63 to 0.82 [13–15]. Moreover, if one technique of the same method for the same sample (e.g., Avicel PH-101) is used, then the CrI value obtained by different researchers turns out to be poorly reproducible [13].

There are several main reasons for the large discrepancies in the values of the crystallinity indices when they are evaluated by different methods or by different measurement techniques of the same method. Firstly, it is due to the application of different calculating equations, mathematical functions, and software to calculate CrI. Secondly, it is due to the different experimental conditions of different methods. Thirdly, it is due to the use of inadequate structural models to evaluate the crystallinity of real cellulose samples. In addition, there are no standard protocols for preparing cellulose samples for the determination of their crystallinity. In particular, the value of the X-ray crystallinity index is affected by the type of crystalline allomorph, the size of crystallites, the distortions, and the texture of the sample [15]. Thus, it is not possible to conclude which of the currently used methods and/or measurement techniques of cellulose crystallinity is the most appropriate.

Therefore, it is advisable to develop a new method operating on other principles, namely the thermochemical method based on the measurement of the wetting enthalpy of various cellulose samples. Unlike the methods discussed above, this thermochemical method is direct, simple, fast, reliable, and reproducible; furthermore, it does not require the use of special models, complex software, and calculations. Several terms are used to express the exothermic effect of the interaction between cellulose and water. This effect is called the "heat" or "enthalpy" of "wetting" or "swelling". In this article, the thermochemical term "enthalpy of wetting" (or "wetting enthalpy") will be used.

The history of the study of the enthalpy of wetting covers more than 90 years. In the early investigations, it was found that the presence of residual moisture in cellulose significantly reduced the exothermic effect of wetting [19]. It was also discovered that the increase in water temperature from 0 to 40 °C reduced the relative value of wetting enthalpy by 14% for the same cellulose sample [20].

More recent studies have shown that the wetting enthalpy of dry and pure samples depends on the structural state of cellulose, in particular, on the ratio between accessible amorphous and inaccessible crystalline domains. For example, it was found that the enthalpy of wetting for cotton fibers is almost twice lower than for viscose fibers, which was explained by the lesser crystallinity of the latter [21].

Estimating the content of accessible amorphous domains in cellulose by the amount of water sorbed in them, a linear correlation between moisture content in cellulose and wetting enthalpy was obtained [22,23]. In addition, a directly proportional relationship was found between the index of non-crystallinity estimated by the WAXS method and the wetting enthalpy for various cellulose samples, both natural and regenerated fibers [23].

These publications show that there is a real opportunity to develop a thermochemical method for determining the degree of amorphicity or its inverse value, the degree of crystallinity of cellulose, which is the main purpose of this study.

## 2. Materials and Methods

### 2.1. Materials

The following cellulose samples were investigated:

➢ Pure chemical-grade cotton cellulose (CC) of Hercules, Inc. (Wilmington, DE, USA).
➢ Microcrystalline cellulose (MCC) prepared by treatment of CC with boiling 2.5 M HCl for 1 h at the acid-to-CC ratio of 20, followed by washing and drying.
➢ Mercerized cotton cellulose (CCM) prepared by treatment of CC with 6 M NaOH for 1 h at room temperature and alkali-to-CC ratio of 20, followed by washing and drying.

➢ Bleached Kraft pine chemical pulp (KP) of Weyerhaeuser, further refined by treatment with 2 M NaOH for 1 h at room temperature and alkali-to-pulp ratio of 20, followed by washing and drying.
➢ Bleached high-pure sulfite spruce pulp (SP) of Weyerhaeuser Co. (Seattle, WA, USA).
➢ Mercerized sulfite pulp (SPM) prepared by treatment of SP with 6 M NaOH for 1 h at room temperature and alkali-to-pulp ratio of 20, followed by washing and drying.
➢ Viscose rayon fibers (VF) of Rayonier, Inc. (Wildlight, FL, USA).

Some characteristics of the used samples are shown in Table 1.

**Table 1.** Characteristics of cellulose samples.

| Sample | *CrA | $\alpha$-Cellulose, % | DP | CrI |
|--------|------|----------------------|-----|-----|
| CC | CI | 98 ± 0.3 | 2700 ± 120 | 0.83 ± 0.02 |
| MCC | CI | 88 ± 0.5 | 170 ± 30 | 0.92 ± 0.01 |
| CCM | CII | 99 ± 0.2 | 2100 ± 110 | 0.68 ± 0.02 |
| KP | CI | 97 ± 0.2 | 1200 ± 80 | 0.78 ± 0.02 |
| SP | CI | 94 ± 0.4 | 1100 ± 90 | 0.75 ± 0.03 |
| SPM | CII | 98 ± 0.3 | 960 ± 60 | 0.66 ± 0.03 |
| VF | CII | - | 250 ± 30 | 0.54 ± 0.02 |

*CrA denotes the main type of crystalline allomorph of cellulose sample.

### 2.2. Methods

#### 2.2.1. Characterization

The content of $\alpha$-cellulose was tested according to the standard TAPPI method T-203 [24]. The average degree of polymerization (DP) was calculated from the viscosity of diluted cellulose solutions in Cuen [25]. The CrI of the samples was estimated using the peak heights measurement technique of WAXS [13,14,26]. Moreover, for some samples (e.g., CC and VF), such characteristics as specific volume, specific gravity, sorption of water vapor and alkali, accessibility for deuterium, and hydrolyzability were also measured according to methods described in the references [2,3,8,27,28].

#### 2.2.2. WAXS

In the WAXS method, the experiments were carried out on a Rigaku-Ultima Plus diffractometer (CuK$_\alpha$—radiation, $\lambda$ = 0.15418 nm) in the 2$\Theta$-angle range from 5 to 50° using a reflection mode. Collimation included a system consisting of vertical slits and Soller slits. The procedure of 0.02° step-by-step scanning was used to determine the exact position of the peaks. The tested specimens in the shape of tablets with a diameter of 16 mm and a thickness of 2 mm were prepared by pressing crushed cellulose samples in a mold at a pressure of 50 MPa. To estimate the CrI, the peak heights measurement technique was used. The calculation was made according to the equation [26]:

$$\text{CrI} = 1 - (I_{min}/I_{max}) \tag{1}$$

where $I_{min}$ is the minimum intensity of X-ray diffraction at 2$\Theta$ = 18° for samples of cellulose I and at 2$\Theta$ = 15° for samples of cellulose II, while $I_{max}$ is the maximum intensity of the (200) peak at 2$\Theta$ = 22–23° for samples of cellulose I and of the (110) peak at 2$\Theta$ = 20–21° for samples of cellulose II.

Three diffractograms were recorded for each cellulose sample to calculate the average CrI value and its standard deviation. The obtained results are shown in Table 1.

#### 2.2.3. Sorption of Water Vapor

The sorption experiments were carried out at 298 K in a vacuum Mac-Ben apparatus having helical spring quartz scales (accuracy ±0.5 mg). Sorption isotherms were obtained in the range of relative pressure of water vapor from 0 to 0.9. Prior to starting the experiments, the samples were dried at 378 K in a vacuum chamber up to constant weight and

additionally dried and degassed in the sorption device. Three of the same samples were tested to calculate an average sorption value and standard deviation.

### 2.2.4. Enthalpy of Wetting

The enthalpy of cellulose wetting with water ($\Delta H_w$) was studied at 298 K using a TAM Precision Solution Calorimeter [29]. Small cellulose samples were used in the form of pieces, fibers, or powders. Prior to starting the experiments, the air-dry sample was weighed into a special glass ampoule and dried in a vacuum at 378 K to a constant weight. The glass ampoule containing the dry sample was sealed and introduced into the calorimetric cell filled with distilled water. The calorimeter was thermostated at 298 K to achieve an equilibrium state. Thereafter, the sealed ampoule with the dry sample was broken to ensure that the cellulose sample was wetted with water. The released exothermic heat effect was measured with accuracy ±0.01 J. Three of the same sample were tested to calculate an average enthalpy value and standard deviation.

### 3. Results

The X-ray diffraction pattern of the cotton cellulose (CC) was typical of natural cellulose containing a crystalline allomorph of CI with characteristic peaks at 2Θ angles of 14.9, 16.5, 22.7, and 34.7° due to X-ray diffraction from planes of the CI-crystalline lattice with Miller indices of ($1\bar{1}0$), (110), (200), and (004), respectively [30] (Figure 1, X-pattern 1).

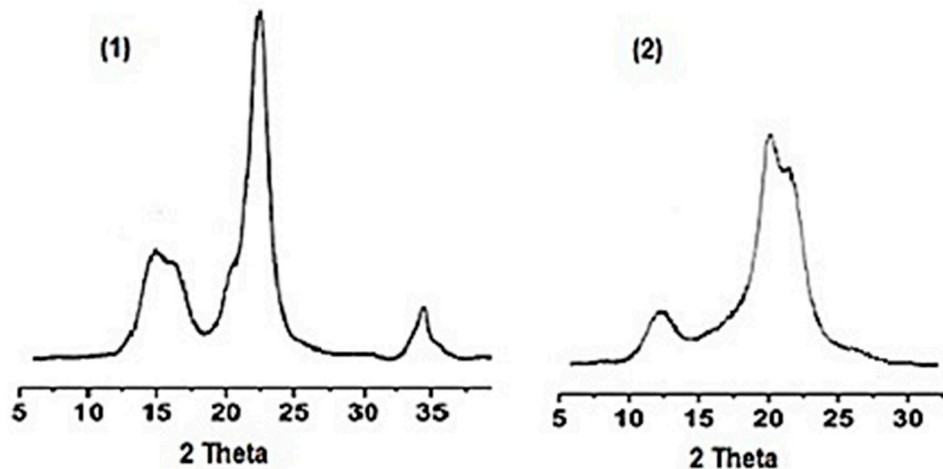

**Figure 1.** X-ray patterns of original (**1**) and mercerized (**2**) CC.

The diffraction patterns of the isolated and bleached wood pulps (KP and SP) were similar to those of cotton cellulose, with the difference that the intensity of the peaks was slightly less and their width was somewhat larger. As a result, the samples of wood pulps showed a lesser CrI value than CC (Table 1). On the other hand, acid hydrolysis of cotton cellulose to obtain MCC led to an enhancement in the intensity of the CI peaks and an increase in CrI value due to the partial removal of the amorphous fraction from the original CC during hydrolysis (Figure 2, X-ray pattern 1).

After the mercerization of cotton cellulose or sulfite pulp, the diffraction pattern of CI allomorph turned into a diffraction pattern characteristic of CII allomorph, which contains peaks at 2Θ angles of 12.4, 20.5, 22.0, and also 34.6° [31] (Figure 1, X-pattern 2). In addition, the intensity of these peaks was lower than the peaks of the original CI samples, which is caused by partial decrystallization during mercerization. The least structurally ordered was a sample of viscose rayon fibers, the CII diffraction pattern of which had low intensive and broad peaks (Figure 2, X-ray pattern 2).

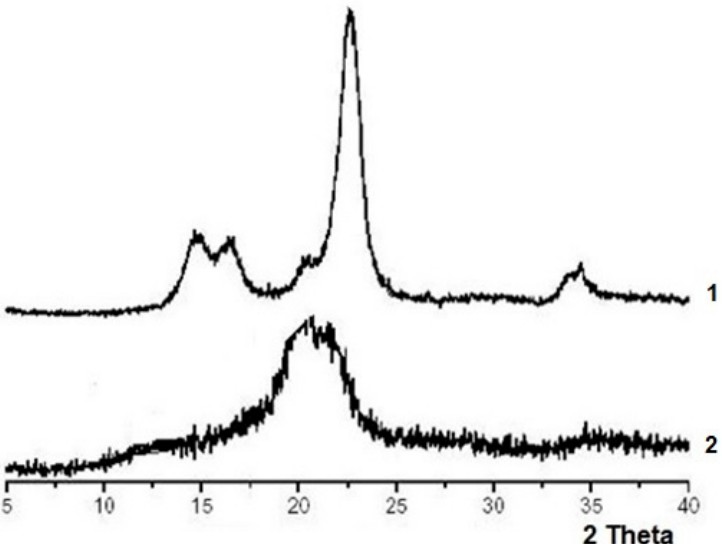

**Figure 2.** X-ray patterns of MCC (1) and VF (2).

Thus, the comparative analysis showed that the MCC sample was the most crystalline and the VF sample was the most amorphous. However, this comparative assessment is not enough to judge the true crystallinity degree of cellulose samples. For this purpose, special studies were carried out, described below.

As is known from chemical thermodynamics, liquid water and equilibrium saturated water vapor have the same chemical potential. This means that such an absorbent as cellulose will absorb the same equilibrium amount of water from the liquid phase as from the saturated vapor phase, having relative vapor pressure $P/P_o = 1$. It is also known that cellulose crystallites are inaccessible to water molecules and their absorption occurs only by amorphous domains of this biopolymer [21–23].

Moreover, the interaction of cellulose with liquid water is accompanied by an exothermic thermal effect, such as wetting enthalpy [15,21–23]. Thus, the lower the crystallinity degree ($X$) and the higher the amorphicity degree ($Y = 1 - X$), the greater will be the absorption value ($A_o$) of saturated water vapor and the higher will be the exothermic wetting enthalpy ($\Delta H_w$) of the cellulose sample. This provides a linear correlation, $\Delta H_w = k\,A_o$, from which one can find also the maximum wetting enthalpy value ($\Delta H_{w,a}$) for completely amorphous cellulose. Then, the crystallinity degree ($X$) expressed in mass fractions for various cellulose samples can be calculated by the equation:

$$X = 1 - (\Delta H_w / \Delta H_{w,a}) \tag{2}$$

To implement such an algorithm, it was necessary to determine the $A_o$ value for various cellulose samples. For this purpose, sorption isotherms of water vapor (WV) were studied in a wide range of relative vapor pressures, $P/P_o$. It was found that these isotherms for various cellulose materials have a sigmoid shape and belong to type II (see, e.g., Figure 3). Isotherms of this type have an initial steep stage, after which a gradual and then accelerated increase in sorption is observed, and, finally, at high relative vapor pressures, a sharp rise in the sorption of WV occurs.

Along with the experimental isotherms, Figure 3 shows also the theoretical isotherm 6 for completely amorphous cellulose (AC), calculated by the method of the additive contributions of hydroxyl groups of amorphous polymers to the sorption of water molecules proposed by Van Krevelen [32].

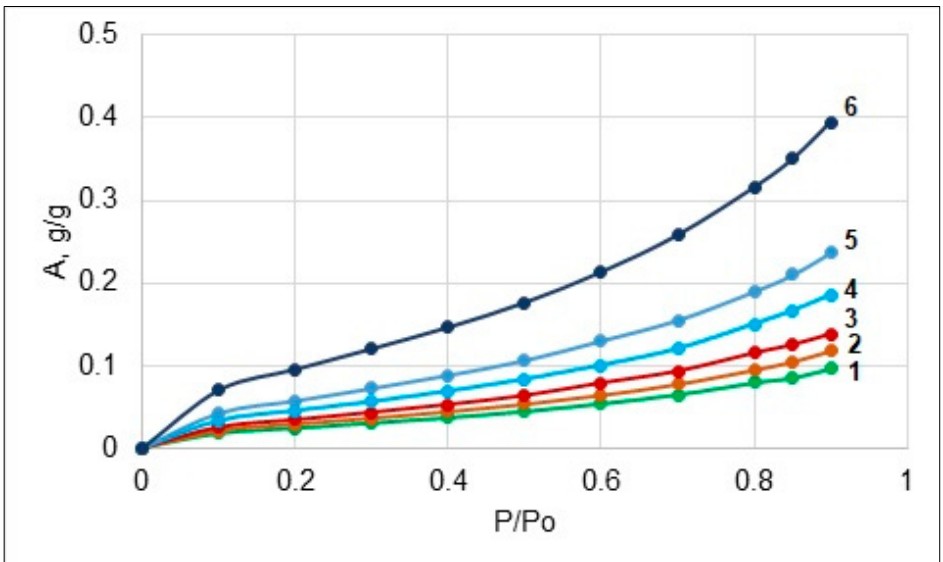

**Figure 3.** Sorption isotherms of water vapor (WV) for samples of MCC (1), CC (2), SP (3), SPM (4), VF (5), and AC (6).

Despite the complex sigmoid shape, such isotherms can be presented in a linear form (Figure 4), using the following equation [33]:

$$A^{-1} = A_o^{-1} - K \cdot \ln(P/P_o) \tag{3}$$

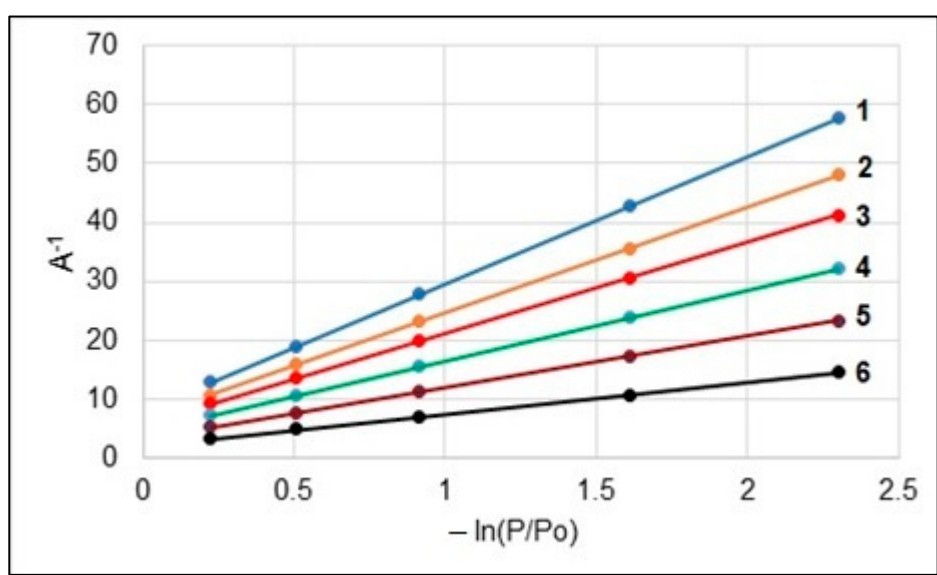

**Figure 4.** Linear form of sorption isotherms for samples of MCC (1), CC (2), SP (3), SPM (4), VF (5), and AC (6).

Extrapolation of the linear plot $A = F\{(-\ln(P/P_o)\}$ to $\ln(P/P_o) = 0$ gives the value of $A_o^{-1}$, from which the maximum amount of water, $A_o$, absorbed by a sample from saturated water vapor at 298 K, can be found (Table 2).

Thermochemical experiments showed that the wetting enthalpy of studied cellulose samples ranged from $-42.2$ to $-104.2$ J/g (Table 2). Moreover, the absolute value of wetting enthalpy is directly proportional to the maximum value of moisture absorption (Figure 5).

**Table 2.** Maximum moisture absorption and wetting enthalpy for cellulose samples.

| Sample | $A_o$, g/g | $-\Delta H_w$, J/g |
|--------|------------|-------------------|
| CC | $0.144 \pm 0.002$ | $46.5 \pm 0.2$ |
| MCC | $0.125 \pm 0.003$ | $42.2 \pm 0.2$ |
| CCM | $0.226 \pm 0.005$ | $75.6 \pm 0.5$ |
| KP | $0.174 \pm 0.003$ | $58.5 \pm 0.3$ |
| SP | $0.183 \pm 0.004$ | $62.1 \pm 0.4$ |
| SPM | $0.236 \pm 0.005$ | $78.9 \pm 0.3$ |
| VF | $0.310 \pm 0.004$ | $104.2 \pm 0.5$ |
| AC | 0.500 * | 168 * |

* Calculated values.

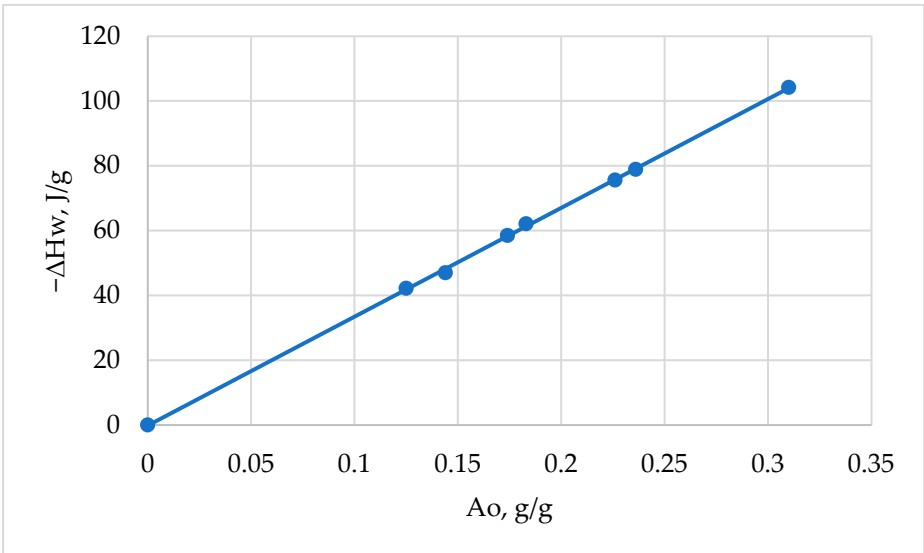

**Figure 5.** Dependence of wetting enthalpy on maximum moisture absorption value for various cellulose samples.

The obtained results are consistent with the literature data. For example, for cotton cellulose, $A_o$ was in the range of 0.14–0.20 g/g and $\Delta H_w$ in the range from $-45$ to $-48$ J/g, while, for viscose rayon fibers, $A_o$ was in the range of 0.27–0.32 g/g and $\Delta H_w$ in the range from $-100$ to $-105$ J/g [21–23].

## 4. Discussion

From Figure 5, it follows that the wetting enthalpy is a linear function of maximum absorption value:

$$\Delta H_w = k \cdot A_o \qquad (4)$$

where coefficient $k = -336$ J/g.

Then, using the maximum moisture absorption value, $A_{o,a} = 0.5$ (g/g), found for completely amorphous cellulose (Table 2), one can calculate the numerical value of the maximum wetting enthalpy, $\Delta H_{w,a}$, for such an amorphous cellulose sample, as follows:

$$\Delta H_{w,a} = k \cdot A_{o,a} = -336 \times 0.5 = -168 \text{ (J/g)} \qquad (5)$$

It should be noted that the direct experimental determination of the numerical value of $\Delta H_{w,a}$ is impossible due to the crystallization of amorphous cellulose during contact with water [34,35], which distorts the true value of the maximum wetting enthalpy.

Substituting the obtained $\Delta H_{w,a}$ value ($-168$ J/g) into Equation (2), one can calculate the thermochemical degree of crystallinity (X) for any cellulose sample (Table 3). In addition, the degree of cellulose amorphicity (Y) can be also calculated, as follows:

$$Y = 1 - X = \Delta H / \Delta H_{w,a} \tag{6}$$

**Table 3.** Degree and index of crystallinity and degree of amorphicity for cellulose samples.

| Sample | Y | X | CrI | RD, % |
|--------|------|------|------|------|
| CC | $0.28 \pm 0.01$ | $0.72 \pm 0.01$ | $0.83 \pm 0.02$ | 15 |
| MCC | $0.25 \pm 0.01$ | $0.75 \pm 0.01$ | $0.92 \pm 0.01$ | 23 |
| CCM | $0.45 \pm 0.01$ | $0.55 \pm 0.01$ | $0.68 \pm 0.02$ | 24 |
| KP | $0.35 \pm 0.01$ | $0.65 \pm 0.01$ | $0.78 \pm 0.02$ | 20 |
| SP | $0.37 \pm 0.01$ | $0.63 \pm 0.01$ | $0.75 \pm 0.03$ | 19 |
| SPM | $0.47 \pm 0.01$ | $0.53 \pm 0.01$ | $0.66 \pm 0.03$ | 24 |
| VF | $0.62 \pm 0.01$ | $0.38 \pm 0.01$ | $0.54 \pm 0.02$ | 42 |

Using the literature data on the wetting enthalpy of CC and VF samples [21–23], it can be calculated by Equation (2) that the X value of these samples is $0.72 \pm 0.01$ and $0.39 \pm 0.01$, respectively. These independent studies confirm the X values of the cellulose samples obtained in this research, which indicates the reliability and reproducibility of the thermochemical method.

As can be seen from the results, the thermochemical method provides the determination of the crystallinity degree of cellulose with a standard deviation (SD) of no more than $\pm 0.01$. In contrast to the thermochemical method, the existing methods used currently for assessing the cellulose crystallinity by CrI, such as WAXS, solid-state [13]C NMR, FTIR, Raman spectroscopy, etc., have 3 to 10 times greater SD values. In addition, the CrI value is poorly reproducible.

A comparison of the determined X values with the CrI values for various cellulose samples showed that the relative deviation (RD) of the CrI value from the real X value is large, from 15% for CC to 42% for VF (Table 3). Ternite and coauthors found [14] that values of CrI for cotton cellulose obtained by means of different measurement techniques of WAXS can vary in the wide range; using these data, it can be calculated that the average relative deviation of CrI from the real degree of crystallinity is 15%. Such large discrepancies indicate that such a parameter as the crystallinity index estimated by different methods and techniques, as a rule, cannot characterize the real content of crystallites in cellulose samples. In some cases, CrI measured by the same method and technique can be used only for a comparative assessment of the crystallinity of cellulose samples.

On the other hand, the obtained degree of crystallinity or amorphicity describes the real supramolecular structure and provides a prediction of various important characteristics of cellulose samples. It is known that many physical, physicochemical, and chemical properties of crystalline and amorphous cellulose differ significantly [2,3,27,36]. For example, the specific volume of amorphous cellulose is higher and the specific gravity is lower than that of crystalline cellulose. The sorption capacity of amorphous cellulose for vapors, molecules, and ions of various substances is relatively high, while that of crystalline cellulose is low and may even be zero. In addition, amorphous cellulose is highly reactive, while the theoretical reactivity of crystalline cellulose should be low. In practice, various cellulose samples are semi-crystalline polymers, so their properties (Z) depend on the degree of crystallinity (X), as follows:

$$Z = X \cdot Z_c + (1 - X) \, Z_a = Z_a - X \, (Z_a - Z_c) \tag{7}$$

where $Z_c$ and $Z_a$ are the properties of completely crystalline ($X = 1$) and completely amorphous ($X = 0$) cellulose, respectively (Table 4).

**Table 4.** Some properties of crystalline and amorphous cellulose [2,3].

| Properties | Symbol | Zc | Za |
|---|---|---|---|
| Specific volume, $cm^3/g$ | Vs | 0.617 | 0.694 |
| Moisture content (%) at standard RH = 65% | Aw | 0 | 23 |
| Accessibility for deuterium | $A_D$ | 0.14 | 1.14 |
| Sorption of alkali from 1 M NaOH, mmol/g | $A_A$ | 0 | 3.1 |
| Hydrolysability (%) after treatment with boiling 2.5 M HCl, 1 h | H | 0 | 40 |

Using Equation (7), it is possible to calculate various properties of cellulose materials, such as specific volume, specific gravity, amount of sorbed moisture from the vapor phase, amount of sorbed alkalis from aqueous solutions, accessibility for deuterium, the acidic hydrolysability, as well as amount of sorbed iodine and dyes from aqueous solutions, enzymatic digestibility, coefficient of thermal expansion, specific heat capacity, etc. [2,3,8]. To illustrate the possibility to predict some properties of cellulose materials, two samples, namely cotton cellulose (CC) and viscose rayon fibers (VF) having a high (X = 0.72) and low (X = 0.38) degree of crystallinity, respectively, were chosen. The properties of these samples were calculated (Calc) by Equation (7), after which they were compared with the experimentally obtained properties (Exp).

As can be seen from Table 5, the calculations are in good agreement with the experimental data, which confirms the possibility of predicting the various characteristics of cellulose materials using such a parameter as real crystallinity degree.

**Table 5.** Some properties of CC and VF.

| Properties | CC | | VF | |
|---|---|---|---|---|
| | **Calc** | **Exp** | **Calc** | **Exp** |
| Vs, $cm^3/g$ | 0.638 | 0.640 | 0.665 | 0.661 |
| *Gs, $g/cm^3$ | 1.57 | 1.56 | 1.50 | 1.51 |
| Aw, % | 6.4 | 6.5 | 14.3 | 14.1 |
| $A_D$ | 0.42 | 0.43 | 0.76 | 0.75 |
| $A_A$, mmol/g | 0.87 | 0.86 | 1.92 | 1.94 |
| H, % | 11 | 10 | 25 | 26 |

*Gs = 1/Vs denotes the specific gravity.

It can be noted that the thermochemical method for determining the crystallinity degree of cellulose samples is direct, simple, fast, precise, reliable, and reproducible. To measure the wetting enthalpy, any type of precise calorimeter produced by high-tech companies can be applied, such as adiabatic, isothermal, microcalorimeter, etc. Furthermore, it does not require the use of special models, complex software, and calculations. In this method, there are no special requirements for the shape and size of the samples. They do not need to be pressed or crushed. It is possible to use cellulose samples with different morphology and type of crystalline allomorph (CI, CII, CIII, or CIV) in the form of pieces, fibers, or powders.

There are only two main conditions to prepare cellulose samples for testing, namely, they must be chemically pure and completely dry. For the removal of moisture from samples, a conventional vacuum drying at 378 K to constant weight can be used.

## 5. Conclusions

In this study, a thermochemical method for determining the real degree of crystallinity (X) of various cellulose samples based on the measurement of their wetting enthalpy ($\Delta H_w$) was proposed. For this purpose, the equation for calculating the X value was derived, as follows: $X = 1 - (\Delta H_w / \Delta H_{w,a})$, where $\Delta H_{w,a} = -168$ (J/g) is wetting enthalpy of completely amorphous cellulose. It was found that the sample of viscose rayon fibers has a minimum X = 0.38, while the MCC sample has a maximum X = 0.75.

A comparison of the determined degree of crystallinity (X) with the crystallinity index for (CrI) various cellulose samples showed that the relative deviation of CrI from the real X ranges from 15 to 42%. Such large discrepancies indicate that such an approximate parameter as the CrI cannot characterize the real content of crystallites in cellulose samples. On the other hand, the obtained degree of crystallinity or amorphicity describes the real supramolecular structure of cellulose and can be used for the prediction of important properties of cellulose samples, such as specific volume, specific gravity, amount of sorbed moisture from the vapor phase, amount of sorbed alkalis from aqueous solutions, accessibility for deuterium, acidic hydrolysability, etc.

**Funding:** This research received no external funding.

**Institutional Review Board Statement:** Not applicable.

**Informed Consent Statement:** Not applicable.

**Data Availability Statement:** Data can be obtained from the corresponding author upon reasonable request.

**Conflicts of Interest:** The author declares no conflict of interest.

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
