# Peer review of "Application of Thermochemical Method to Determine the Crystallinity Degree of Cellulose Materials"

_applsci, doi:10.3390/app13042387_

Round 1

Reviewer 1 Report

This paper introduced an Application of thermochemical method to determine the crystallinity degree of cellulose materials. The author has done a lot of work. I think that this paper can be published. There are many weaknesses, so it is better to revise them carefully.

1. Line 57: there is an absence of space in the first sentence.

2. Line 61: please check “thirdly”.

3. Line 211: there is an absence of space in the first sentence.

4. Line 282: there is a large space in the first sentence.

5. References: please check “8” and “15”.

Author Response

All comments of the reviewer were taken into account by the author and the corresponding corrections were made.

Reviewer 2 Report

The author used the thermochemical method for determining the real degree of crystallinity (X) of cellulose based on the measurement of the enthalpy of wetting.

This research report has a novelty and good contribution to the literature.

Author Response

Conclusions were improved

Author Response

Conclusions were improved
